# BNPO: Beta Normalization Policy Optimization

## Abstract

Recent studies, including DeepSeek-R1 and Kimi-k1.5, have demonstrated that reinforcement learning with rule-based, binary-valued reward functions can significantly enhance the reasoning capabilities of large language models. These models primarily utilize REINFORCE-based policy optimization techniques, such as REINFORCE with baseline and group relative policy optimization (GRPO). However, a key limitation remains: current policy optimization methods either neglect reward normalization or employ static normalization strategies, which fail to adapt to the dynamic nature of policy updates during training. This may result in unstable gradient estimates and hinder training stability. To address this issue, we propose Beta Normalization Policy Optimization (BNPO), a novel policy optimization method that adaptively normalizes rewards using a Beta distribution with dynamically updated parameters. BNPO aligns the normalization with the changing policy distribution, enabling more precise and lower-variance gradient estimation, which in turn promotes stable training dynamics. We provide theoretical analysis demonstrating BNPO's variance-reducing properties and show that it generalizes both REINFORCE and GRPO under binary-valued reward settings. Furthermore, we introduce an advantage decomposition mechanism to extend BNPO's applicability to more complex reward systems. Experimental results confirm that BNPO achieves state-of-the-art performance among policy optimization methods on reasoning tasks.

## 1 Introduction

Kimi-K1.5 (Team et al., 2025) and DeepSeek-R1 (Guo et al., 2025) have demonstrated that reinforcement learning can substantially enhance the reasoning capabilities of large language models. These models leverage reinforcement learning techniques built on rule-based, binary-valued outcome reward functions, and utilize policy optimization techniques such as REINFORCE with baseline (Kool et al., 2019) and group relative policy optimization (GRPO) (Shao et al., 2024).

In contrast to proximal policy optimization (PPO) (Schulman et al., 2017), which employs a critic network to estimate the baseline for policy gradients, REINFORCE with baseline and GRPO utilize Monte Carlo sampling for baseline estimation, reducing memory and computational overhead. Specifically, REINFORCE with baseline incorporates a state-dependent baseline compared to vanilla REINFORCE to reduce gradient variance, while GRPO further stabilizes training by normalizing rewards, thereby reducing gradient variance in high-variance reward scenarios.

Despite these advances, a fundamental limitation remains: current methods either lack reward normalization entirely or use fixed normalization terms throughout training. This is suboptimal, as the policy model evolves during training, fixed normalization cannot adapt to such dynamics, potentially resulting in inaccurate gradient estimates and unstable learning.

To overcome this limitation, we propose a novel policy optimization method, called Beta Normalization Policy Optimization (BNPO), which dynamically normalizes the reward function using a Beta distribution with its adaptive parameters. By evolving alongside the policy model, this normalization mechanism provides more accurate and lower-variance gradient estimates. Besides, we introduce an advantage decomposition mechanism to enhance BNPO's ability to handle complex reward systems.

Our approach is motivated by the observation that, under binary-valued reward functions, the reward can be seen as a random variable with Bernoulli distribution, and its expectation naturally can be modeled as a random variable with Beta distribution. As training progresses and the policy evolves,

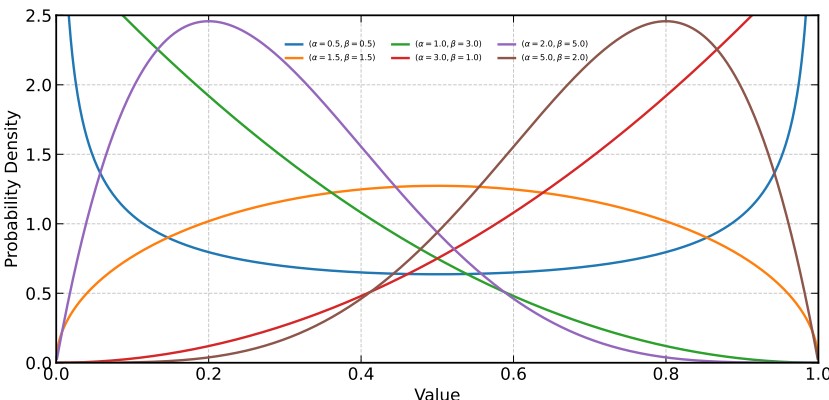

Figure 1: Probability density function of Beta distribution.

the distribution of expected rewards also shifts. BNPO explicitly accounts for these shifts by adjusting the normalization term accordingly.

We further present a theoretical analysis demonstrating that BNPO can effectively reduce the variance of policy gradient estimates when the Beta distribution parameters are appropriately set. Moreover, we show that BNPO generalizes both REINFORCE and GRPO in the binary-valued reward setting, highlighting its broad applicability and theoretical consistency. Finally, experimental results show that BNPO achieves state-of-the-art performance in policy optimization for reasoning tasks.

## 2 BACKGROUND

### 2.1 BETA DISTRIBUTION

The Beta distribution is a continuous probability distribution defined on the interval $[0, 1]$, making it particularly well-suited for modeling probabilities. In this paper, we use it to represent the distribution of the expectation of a binary-valued reward. Its probability density function is given by

$$f(p; \alpha, \beta) = \frac{1}{\mathrm{B}(\alpha, \beta)} p^{\alpha-1}(1-p)^{\beta-1}, \quad p \in [0, 1] \text{ or } p \in (0, 1), \quad \alpha > 0, \beta > 0, \tag{1}$$

where $\mathrm{B}(\cdot, \cdot)$ denotes the Beta function, which serves as a normalization constant to ensure the probability density function integrates to one. Figure 1 illustrates the probability density function of the Beta distribution under various parameter settings.

The shape of the Beta distribution is primarily determined by the values of $\alpha$ and $\beta$, which control the concentration of probability mass and the skewness of the distribution. When $\alpha > 1$ and $\beta > 1$, the distribution is unimodal and bell-shaped, with the mode $\frac{\alpha-1}{\alpha+\beta-2}$ lying between 0 and 1. If $\alpha < 1$ and $\beta < 1$, the distribution becomes U-shaped, with higher densities near 0 and 1. When one of the parameters is less than 1 while the other is greater than 1, the distribution becomes highly skewed, concentrating mass near one endpoint. A special case occurs when $\alpha = \beta$, resulting in a symmetric distribution centered around $p = \frac{1}{2}$. This adaptability makes the Beta distribution a popular choice in probabilistic modeling contexts.

In terms of summary statistics, the mean of the Beta distribution is given by $\mathbb{E}[p] = \frac{\alpha}{\alpha+\beta}$, reflecting the balance between the two parameters. The variance is given by $\mathrm{Var}[p] = \frac{\alpha\beta}{(\alpha+\beta)^2(\alpha+\beta+1)}$, which decreases as the sum $\alpha + \beta$ increases, indicating greater certainty or concentration around the mean.

### 2.2 POLICY OPTIMIZATION

Reinforcement learning provides an effective framework for training large language models by enabling them to learn policies through interaction with the environment and feedback signals. Among

various reinforcement learning methods, policy gradient techniques are particularly prominent due to their ability to scale to high-dimensional action spaces typical in language generation tasks. Given a outcome reward function $R(q, o)$, the objective function in policy gradient methods is defined as:

$$\mathcal{L}(\theta) = \mathbb{E}_{q \sim \rho, \, o \sim \pi_\theta(\cdot|q)}\Big[R(q, o)\Big], \tag{2}$$

where $\rho$ represents the distribution of questions $q$, and $\pi_\theta(\cdot|q)$ denotes the parameterized policy model that defines the distribution over outputs $o$. According to the policy gradient theorem (Sutton et al., 1999), the policy gradient for the objective in Eq.(2) is given by:

$$\nabla_\theta \mathcal{L}(\theta) = \mathbb{E}_{q \sim \rho, \, o \sim \pi_\theta(\cdot|q)}\Big[\nabla_\theta \log \pi_\theta(o|q)\, R(q, o)\Big]. \tag{3}$$

In practice, directly using Eq. (3) can lead to high variance in gradient estimates (Barto, 2021), which negatively impacts training stability. To mitigate this, policy gradient methods typically introduce an advantage function $A(q, o)$:

$$\nabla_\theta \mathcal{J}(\theta) = \mathbb{E}_{q \sim \rho, \, o \sim \pi_\theta(\cdot|q)}\Big[\nabla_\theta \log \pi_\theta(o|q)\, A(q, o)\Big], \tag{4}$$

where $A(q, o)$ represents the relative advantage of a question-output pair $(q, o)$ compared to other pairs. The use of $A(q, o)$ primarily serves to reduce the variance in policy gradient estimation:

$$\text{Var}_{q \sim \rho, \, o \sim \pi_\theta(\cdot|q)}\Big[\nabla_\theta \log \pi_\theta(o|q)\, A(q, o)\Big]. \tag{5}$$

The advantage function is commonly formulated as $A(q, o) = \frac{R(q,o) - \mu}{\sigma}$, where $\mu$ serves as a baseline for $R(q, o)$ to compare and $\sigma$ acts as a normalization term. With appropriate choices of $\mu$ and $\sigma$, the estimation of policy gradient remains unbiased while its variance is reduced.

REINFORCE with baseline (Team et al., 2025; Kool et al., 2019) defines $A(q, o)$ as

$$\begin{aligned} A(q, o) =& R(q, o) - \mathbb{E}_{o' \sim \pi_\theta(\cdot|q)}\Big[R(q, o')\Big] \\ \approx& R(q, o) - \text{Mean}(\{R(q, o'_j)\}_{j=1}^m), \end{aligned} \tag{6}$$

where the baseline is the mean reward over a sampled group of outputs $\{(q, o'_j)\}_{j=1}^m$. This Monte Carlo estimate approximates the expected reward $\mathbb{E}_{o' \sim \pi_\theta(\cdot|q)}\Big[R(q, o')\Big]$ and has been shown to effectively reduce the variance of policy gradient estimates (Wu et al., 2018).

GRPO (Guo et al., 2025; Shao et al., 2024) defines $A(q, o)$ as:

$$\begin{aligned} A(q, o) =& \frac{R(q, o) - \mathbb{E}_{o' \sim \pi_\theta(\cdot|q)}\Big[R(q, o')\Big]}{\sqrt{\text{Var}_{o' \sim \pi_\theta(\cdot|q)}[R(q, o')]}} \\ \approx& \frac{R(q, o) - \text{Mean}(\{R(q, o'_j)\}_{j=1}^m)}{\sqrt{\text{Var}(\{R(q, o'_j)\}_{j=1}^m)}}, \end{aligned} \tag{7}$$

Compared to REINFORCE with baseline, GRPO further uses the standard deviation of the rewards over the sampled set $\{(q, o'_j)\}_{j=1}^m$ to normalize the reward function. This normalization term can further reduce the variance in estimating policy gradient for high-variance reward functions.

PPO (Schulman et al., 2017) further enhances stability by incorporating importance sampling and a clipping mechanism for off-policy updates:

$$\mathcal{L}(\theta) = \mathbb{E}_{q \sim \rho, o \sim \pi_{\theta_{\text{old}}}(o|q)}\Big[\min(\frac{\pi_\theta(o|q)}{\pi_{\theta_{\text{old}}}(o|q)} A(q, o), \text{clip}(\frac{\pi_\theta(o|q)}{\pi_{\theta_{\text{old}}}(o|q)}, 1 - \varepsilon, 1 + \varepsilon) A(q, o))\Big], \tag{8}$$

where $\pi_{\theta_{old}}$ is the old policy and $\varepsilon$ is a hyperparameter that controls the range of clipping.

## 3 BETA NORMALIZATION POLICY OPTIMIZATION

In this section, we introduce our policy optimization method, BNPO, which employs a Beta distribution to normalize binary-valued reward functions. BNPO adapts to the evolving policy model

during training by dynamically adjusting the parameters of the Beta distribution. We then provide a theoretical proof demonstrating that BNPO effectively reduces the variance of policy gradient estimates. Furthermore, we show that BNPO generalizes both REINFORCE with baseline and GRPO in the context of binary-valued rewards. Finally, we present an advantage decomposition mechanism to extend BNPO's applicability to more complex reward systems.

**Beta normalization**  We use the accuracy of an output $o$ with respect to a question $q$ as the reward function $R(q, o)$ as in DeepSeek-R1 (Guo et al., 2025), i.e.,

$$R(q, o) = \begin{cases} 1, & \text{if } o \text{ contains the answer } a \text{ of the question } q, \\ 0, & \text{otherwise.} \end{cases} \tag{9}$$

Since the value of $R(q, o)$ is either 0 or 1, $R(q, o)$ can be treated as a random variable with Bernoulli distribution, i.e.

$$R(q, o) \sim \text{Bernoulli}\,(p(q)), \quad 0 \le p(q) \le 1,$$
$$p(q) = \mathbb{E}_{o \sim \pi_\theta(\cdot|q)}[R(q, o)|q], \tag{10}$$

where $p(q)$ denotes the probability that output $o$ is correct for question $q$, and it is also the expected reward under the distribution $\pi_\theta(\cdot|q)$. As mentioned in Section 2.1, the Beta distribution is very suitable for modeling probability. Thus, we model $p(q)$ as a random variable with Beta distribution $f_D(p(q); a, b)$, where the parameters $a$ and $b$ control the shape of the distribution. These parameters can be estimated using Monte Carlo sampling.

As the policy model $\pi_\theta(\cdot|q)$ evolves during training, the distribution of $p(q)$ also changes dynamically. To account for these changes, we propose using an additional Beta distribution $f_N(p(q); \alpha, \beta)$ to normalize the reward function. The advantage function in our BNPO method is defined as:

$$A_{\alpha,\beta}(q, o) = \frac{R(q, o) - p(q)}{f_N(p(q); \alpha, \beta)}, \tag{11}$$

where $p(q)$ serves as the baseline, as in REINFORCE with baseline and GRPO, and $f_N(p(q); \alpha, \beta)$ is used to normalize the reward function $R(q, o)$.

**The setting of $\alpha$ and $\beta$**  We dynamically adjust the parameters $(\alpha, \beta)$ in $f_N(p(q); \alpha, \beta)$ to ensure that $A_{\alpha,\beta}(q, o)$ adapts to the evolving distribution $f_D(p(q); a, b)$ during training. The primary goal in setting $\alpha$ and $\beta$ is to minimize the variance in policy gradient estimation. We present the following theorem to achieve it.

**Theorem 1.** *Let $q \sim \rho$ be a question and $o \sim \pi_\theta(\cdot|q)$ be an output with reward $R(q, o) \in \{0, 1\}$, where $R(q, o)$ follows a Bernoulli distribution with success probability $p(q) = \mathbb{E}_{o \sim \pi_\theta(\cdot|q)}[R(q, o)|q]$, and that $p(q)$ follows a Beta distribution $f_D(p(q); a, b)$. Define the BNPO gradient estimator as*

$$g_{\alpha,\beta} = \nabla_\theta \log \pi(o|q) \frac{R(q, o) - p(q)}{f_N(p(q); \alpha, \beta)}.$$

*where $f_N(p(q); \alpha, \beta)$ is a Beta distribution. Under the assumption $\nabla_\theta \log \pi(o|q)$ is uncorrelated with $\frac{R(q,o) - p(q)}{f_N(p(q); \alpha, \beta)}$, the variance of the policy gradient estimator $\text{Var}_{q \sim \rho,\, o \sim \pi_\theta(\cdot|q)}(g_{\alpha,\beta})$ is finite if and only if: $\alpha < \frac{a+3}{2}$ and $\beta < \frac{b+3}{2}$. Within this domain, $\text{Var}_{q \sim \rho,\, o \sim \pi_\theta(\cdot|q)}(g_{\alpha,\beta})$ attains a unique minimum at:*

$$\alpha = 1 + \frac{a}{3}, \quad \beta = 1 + \frac{b}{3}.$$

See Appendix C for the proof. The above theorem demonstrates that the optimal parameter settings for minimizing the variance of the policy gradient are $\alpha = 1 + \frac{a}{3}$ and $\beta = 1 + \frac{b}{3}$. Thus, the choice of $(\alpha, \beta)$ depends on the values of $(a, b)$. We estimate $(a, b)$ using the method-of-moments approach and Monte Carlo sampling.

Given the following relationships:

$$\mathbb{E}[p(q)] = \frac{a}{a + b},$$

$$\text{Var}[p(q)] = \frac{ab}{(a + b)^2 (a + b + 1)}, \tag{12}$$

we can solve for $a$ and $b$ as:

$$a = \left( \frac{\mathbb{E}[p(q)](1 - \mathbb{E}[p(q)])}{\mathrm{Var}[p(q)]} - 1 \right) \mathbb{E}[p(q)],$$

$$b = \left( \frac{\mathbb{E}[p(q)](1 - \mathbb{E}[p(q)])}{\mathrm{Var}[p(q)]} - 1 \right) (1 - \mathbb{E}[p(q)]). \tag{13}$$

We then estimate $\mathbb{E}[p(q)]$ and $\mathrm{Var}[p(q)]$ using Monte Carlo methods to get $a$ and $b$:

$$\mathbb{E}[p(q)] \approx \mathrm{Mean}(\{p(q_i)\}_{i=1}^n),$$

$$\mathrm{Var}[p(q)] \approx \mathrm{Var}(\{p(q_i)\}_{i=1}^n). \tag{14}$$

**The interpretation of $\alpha$ and $\beta$** The parameters $\alpha$ and $\beta$ can be understood in terms of the mean and variance of $f_D(p(q); a, b)$. The mean of $f_D(p(q); a, b)$ is given by $\frac{a}{a+b}$, representing the average reward of all $(q, o)$ pairs. The mode of $f_N(p(q); \alpha, \beta)$ is $\frac{\alpha-1}{\alpha+\beta-2} = \frac{a}{a+b}$, which corresponds to the value at which $f_N(p(q); \alpha, \beta)$ attains its maximum. This shows that the mean of $f_D(p(q); a, b)$ is equal to the mode of $f_N(p(q); \alpha, \beta)$. As a result, the reward function $R(q, o)$ is most normalized at the average reward $\frac{a}{a+b}$.

The variance of $f_D(p(q); a, b)$ decreases/increases as the sum $a + b$ increases/decreases. Since $\alpha + \beta = 2 + \frac{a+b}{3}$, the variance of $f_N(p(q); \alpha, \beta)$ behaves similarly: it decreases/increases as $a + b$ increases/decreases. Hence, $f_N(p(q); \alpha, \beta)$ adapts its parameters to align with the variance changes of $f_D(p(q); a, b)$.

**REINFORCE and GRPO** We now demonstrate that BNPO generalizes both REINFORCE with baseline and GRPO under binary-valued reward circumstances, reducing to each of these methods under specific settings for $\alpha$ and $\beta$.

REINFORCE with baseline defines the advantage function $A(q, o)$ as

$$\begin{aligned}
A(q, o) &= R(q, o) - \mathbb{E}_{o' \sim \pi_\theta(\cdot|q)}\Big[R(q, o')\Big] \\
&= R(q, o) - p(q) \\
&= \frac{R(q, o) - p(q)}{f_N(p(q); 1, 1)} \\
&= A_{1,1}(q, o).
\end{aligned} \tag{15}$$

Therefore, BNPO reduces to REINFORCE if $f_N(p(q); \alpha, \beta) = f_N(p(q); 1, 1)$. Since RLOO (Kool et al., 2019; Ahmadian et al., 2024) is equivalent to REINFROCE with baseline up to a scaling constant (Liu et al., 2025), BNPO can also reduce to RLOO.

GRPO defines the advantage function $A(q, o)$ as

$$\begin{aligned}
A(q, o) &= \frac{R(q, o) - \mathbb{E}_{o' \sim \pi_\theta(\cdot|q)}\Big[R(q, o')\Big]}{\sqrt{\mathrm{Var}_{o' \sim \pi_\theta(\cdot|q)}[R(q, o')]}} \\
&= \frac{R(q, o) - p(q)}{\sqrt{p(q)(1 - p(q))}} \\
&\propto \frac{R(q, o) - p(q)}{f_N(p(q); \frac{3}{2}, \frac{3}{2})} \\
&= A_{\frac{3}{2}, \frac{3}{2}}(q, o).
\end{aligned} \tag{16}$$

In training large language models, gradient clipping is commonly employed. Consequently, scaling the loss function by a constant does not affect the parameter update process. Consequently, BNPO reduces to GRPO if $f_N(p(q); \alpha, \beta) = f_N(p(q); \frac{3}{2}, \frac{3}{2})$.

REINFORCE with a baseline and GRPO can be viewed as special cases of BNPO with fixed values of $(\alpha, \beta)$. In contrast, BNPO dynamically adjusts $(\alpha, \beta)$ during training to better align with the evolving policy model.

**Advantage decomposition** To extend our method to more complex reward systems beyond a single binary reward function, we introduce an advantage decomposition mechanism. This approach enables the separate normalization of each individual reward component, leading to a more accurate estimation of the overall advantage function. Such decomposition is particularly beneficial in settings with multiple reward signals. For example, DeepSeek-R1 employs both format and accuracy rewards to ensure that model outputs not only follow the required structure but also produce correct answers.

Given $K$ binary-valued reward functions $\{R^{(1)}(q,o), R^{(2)}(q,o), \cdots, R^{(K)}(q,o)\}$, we decompose the overall advantage function $A(q,o)$ into $K$ sub-advantage functions $A^{(i)}(q,o)$ as follows:

$$A(q,o) = \frac{1}{K} \sum_{i=1}^{K} A^{(i)}(q,o) = \frac{1}{K} \sum_{i=1}^{K} \frac{R^{(i)}(q,o) - p(q)^{(i)}}{f_N(p(q)^{(i)}; \alpha^{(i)}, \beta^{(i)})}. \tag{17}$$

where each sub-advantage function $A^{(i)}(q,o)$ is computed for the corresponding reward function $R^{(i)}(q,o)$.

Unlike previous methods that first sum multiple reward functions and then compute the final advantage function, our approach calculates the advantage function for each individual reward function first, and then averages them to obtain the final advantage function. The key benefit of this approach is that it allows for separate normalization of each reward function, ensuring that the normalization of one function does not interfere with others.

**Extend BNPO to multi-valued or continuous rewards.** We present the detailed implementation of our BNPO in Alg.(1). Although our theoretical analysis of BNPO is based on binary-valued reward functions, the method remains applicable to general reward functions in practice. BNPO can be naturally extended to handle general reward functions by directly assuming that $p(q)$ follows a Beta distribution, without relying on the intermediate assumption that $R(q,o)$ is Bernoulli-distributed. This extension broadens the applicability of BNPO to arbitrary reward types.

In this generalized setting, the solution to Theorem 1, originally derived for binary rewards, serves as an approximate solution. While it may not be strictly optimal, it remains effective in reducing gradient variance.

## 4 RELATED WORK

Reinforcement learning has been widely adopted to align large language models with human preferences, as seen in systems like ChatGPT and DeepSeek-R1. ChatGPT (Ouyang et al., 2022) employs PPO for policy optimization, which relies on a critic network to better estimate policy gradients. However, training a critic network is computationally intensive and memory-demanding, particularly for large language models. To address this, models such as DeepSeek-R1 and Qwen (Yang et al., 2024a) adopt REINFORCE-based methods, which avoid the need for a critic network.

The original REINFORCE algorithm (Williams, 1992) estimates gradients through Monte Carlo sampling but often suffers from high variance, which can hinder learning stability and efficiency. To mitigate this issue, RLOO (Kool et al., 2019) introduces a baseline function that uses the mean reward of a group of samples as a reference, significantly reducing gradient variance, especially when batch sizes are small. ReMax (Li et al., 2024) builds on this idea by employing greedy decoding to obtain a baseline. GRPO (Shao et al., 2024) further refines this idea by normalizing each reward using the standard deviation of the group, reducing variance even more. REINFORCE++ (Hu, 2025) goes a step further by leveraging the rewards of all samples to estimate the policy gradient, resulting in more stable and robust learning performance. However, these methods either lack proper normalization or rely on static normalization strategies, which are insufficient for adapting to the evolving nature of policy during training. In contrast, BNPO dynamically adjusts its normalization parameters in response to changes in the policy, effectively stabilizing training.

Beyond policy optimization methods, normalization techniques are widely used in reinforcement learning. Reward and value normalization have been extensively studied, including adaptive rescaling approaches such as PopArt (Van Hasselt et al., 2016). Large-scale empirical analyses, such as Henderson et al. (2018), highlight the critical role of reward normalization in ensuring stable and reproducible training. In addition, PPO stabilizes training by clipping rewards and policy updates, which acts as

---

**Algorithm 1** BNPO: Beta Normalization Policy Optimization

---

**Input:** Initial policy model $\pi_{\theta_0}$, $K$ binary-valued Reward model $R^{(i)}(q, o)$, training set $\mathcal{D}$,
  number of steps $S$, number of PPO iterations $T$, batch size $n$, number of outputs $m$.
 1: Initialize policy model $\pi_\theta \leftarrow \pi_{\theta_0}$.
 2: **for** step $= 1$ **to** $S$ **do**
 3:  Update the old policy model $\pi_{\theta_{old}} \leftarrow \pi_\theta$.
 4:  Sample $n$ questions $q$ from $\mathcal{D}$.
 5:  Sample $m$ outputs $o \sim \pi_{\theta_{old}}(\cdot|q)$ for each question $q$.
 6:  **for** each question-output pair $(q, o)$ **do**
 7:    **for** $i = 1$ **to** $K$ **do**
 8:      Compute the reward $R^{(i)}(q, o)$.
 9:    **end for**
10:  **end for**
11:  **for** each question $q$ **do**
12:    Estimate $p(q)$ in Eq.(10).
13:  **end for**
14:  Estimate the parameters $a$ and $b$ in $f_D(p(q); a, b)$ by Eq.(13) and Eq.(14).
15:  Set the the parameters $\alpha$ and $\beta$ in $f_N(p(q); \alpha, \beta)$ as $\alpha = 1 + \frac{a}{3}$ and $\beta = 1 + \frac{b}{3}$.
16:  **for** each question-output pair $(q, o)$ **do**
17:    Compute the advantage $A(q, o)$ by Eq.(11) and Eq.(17).
18:  **end for**
19:  **for** iteration $= 1$ **to** $T$ **do**
20:    Update $\pi_\theta$ by maximizing Eq.(8).
21:  **end for**
22: **end for**
**Output:** Optimized policy model $\pi_\theta$.

---

an implicit normalization mechanism to prevent excessively large gradient steps (Schulman et al., 2017). Collectively, these studies demonstrate the broad effectiveness of normalization techniques in reinforcement learning.

## 5 EXPERIMENTS

In this section, we first describe the experimental setup in Section 5.1, followed by the presentation of results in Section 5.2. We then analyze training stability in Section 5.3. We finally show the evolution of the normalization of BNPO in Section 5.4.

### 5.1 EXPERIMENTAL SETTINGS

**Models** To evaluate the effectiveness of BNPO, we conduct experiments on two publicly available base models of different scales: Qwen2.5-Math-1.5B and Qwen2.5-Math-7B (Yang et al., 2024a;b).

**Methods** We compare BNPO method against several policy optimization methods, including REINFORCE, ReMax, GRPO, and REINFORCE++. Since RLOO is equivalent to REINFORCE with a baseline, we only report the results for REINFORCE with baseline.

**Datasets** For training, we utilize the full MATH dataset (Hendrycks et al., 2021), which consists of 7,500 diverse mathematical problems spanning a wide range of topics and difficulty levels.

For evaluation, we use four benchmark datasets: MATH500 (Hendrycks et al., 2021; Lightman et al., 2023), AMC23 (Art of Problem Solving, 2025b), AIME2024 and AIME2025 (Art of Problem Solving, 2025a).

**Metrics** We use pass@1 as the evaluation metric. For the AMC23, AIME 2024, and AIME 2025 datasets, we run the test set 16 times and report the average results, as these test sets are relatively small. We report the best average performance during training.

Table 1: The performance of different policy optimization methods on math datasets.

| Methods | MATH500 | AMC23 | AIME2024 | AIME2025 | Average |
|---|---|---|---|---|---|
| *Qwen2.5-Math-1.5B* | | | | | |
| Base | 28.0 | 27.3 | 6.0 | 3.1 | 16.1 |
| REINFORCE | 72.2 | 53.6 | **18.3** | **11.5** | 38.9 |
| ReMax | 73.2 | 53.3 | 17.1 | 9.6 | 38.3 |
| GRPO | **75.0** | 52.0 | 15.6 | 11.0 | 38.4 |
| REINFORCE++ | 73.8 | 52.0 | 16.7 | 9.8 | 38.1 |
| BNPO | 74.0 | **54.5** | 17.9 | 11.3 | **39.4** |
| *Qwen2.5-Math-7B* | | | | | |
| Base | 41.4 | 32.5 | 11.0 | 5.0 | 22.5 |
| REINFORCE | 78.2 | 65.6 | 32.9 | 11.7 | 47.1 |
| ReMax | 77.8 | 63.6 | **33.5** | **15.4** | 47.6 |
| GRPO | **78.6** | 64.5 | 32.3 | 12.9 | 47.1 |
| REINFORCE++ | **78.6** | 64.4 | 32.1 | 12.3 | 46.8 |
| BNPO | 77.0 | **68.8** | 32.1 | 13.3 | **47.8** |

**Hyperparameters**   For all methods, we set the batch size 32, number of outputs to 16, number of PPO iterations to 1, number of epochs to 5, and learning rate to $10^{-6}$. The temperature is set to 1.0 during training and 0.6 during evaluation. We use the chat template of Qwen2.5-Math-7B and set the maximum question length to 1024 and the maximum output length to 3072, corresponding to the maximum context length of 4096 for Qwen-Math-1.5B and Qwen-Math-7B.

## 5.2 RESULTS

As shown in Table 1, BNPO achieves the highest average performance among all policy optimization methods for both the Qwen2.5-Math-1.5B and Qwen2.5-Math-7B base models, demonstrating its effectiveness and versatility. Notably, BNPO trained on Qwen2.5-Math-7B delivers significant improvements on the AMC23 dataset. In contrast, REINFORCE, GRPO, and REINFORCE++, which either lack normalization or rely on static normalization, exhibit suboptimal performance. Although ReMax achieves performance comparable to BNPO on the Qwen2.5-Math-7B model, it requires additional sampling to dynamically estimate the baseline, resulting in approximately 25% longer training times in our experiments.

## 5.3 TRAINING STABILITY

We have demonstrated in Theorem 1 that BNPO effectively reduces gradient variance, thereby enhancing training stability. Given the substantial computational cost of training large language models, maintaining stable training dynamics is crucial. To evaluate this, we use the gradient norm, an indicator of policy variance, as a proxy for training stability.

As shown in Figure 2, BNPO exhibits the highest stability among the methods, with consistently stable gradient norms throughout training. In contrast, GRPO, REINFORCE, Remax and REINFORCE++ show more significant fluctuations, indicating less stable training. These results highlight the benefit of BNPO's dynamic normalization mechanism over the static normalization used in GRPO and REINFORCE++.

## 5.4 EVOLUTION OF NORMALIZATION

Our BNPO method dynamically adjusts the parameters $(\alpha, \beta)$ in the normalization $f_N(p(q); \alpha, \beta)$ to ensure that the advantage function $A_{\alpha,\beta}(q, o)$ remains aligned with the evolving expected reward distribution $f_D(p(q); a, b)$ throughout training. We have further provided an interpretation of $\alpha$ and $\beta$ in terms of the mean and variance of $f_D(p(q); a, b)$, demonstrating how they can be related to $\mathbb{E}[p(q)]$ and $\text{Var}[p(q)]$.

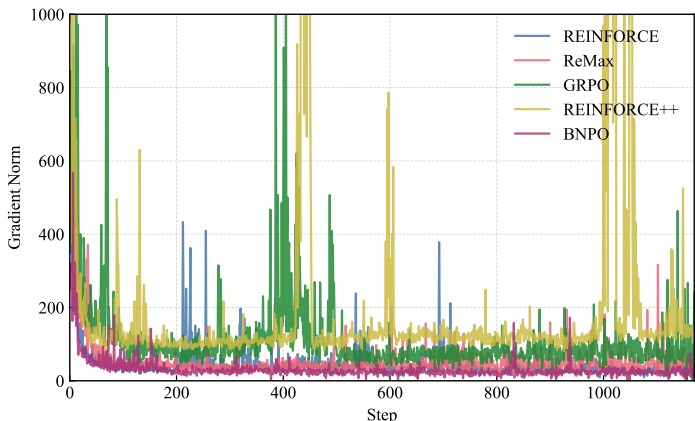

Figure 2: The norm of gradient during training.

To illustrate this relationship, Figure 3 presents the evolution of $(\mathbb{E}[p(q)], \mathrm{Var}[p(q)], \alpha, \beta)$ over the course of training. For clarity, we recommend focusing on $\alpha$ (depicted in green), which exhibits clear variations throughout the training process. These fluctuations in $\alpha$ demonstrate that BNPO actively modifies its distributional parameters in response to changes in the expectation and variance of $p(q)$. This adaptive behavior helps reduce gradient variance, thereby contributing to the stability and effectiveness of the training process.

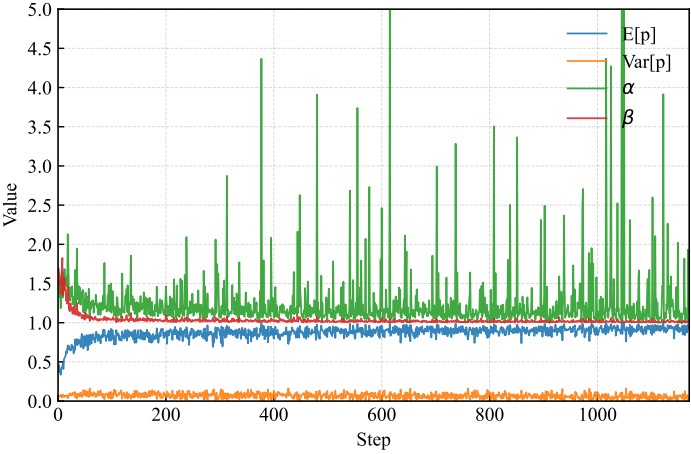

Figure 3: The values of $(\mathbb{E}[p(q)], \mathrm{Var}[p(q)], \alpha, \beta)$ during training.

## 6 CONCLUSION

In this paper, we propose a new policy optimization methods, BNPO, which use Beta distribution to normalize the reward function. We find that the expectation of a binary-valued reward function can be treated as a random variable with Beta distribution, thus, we use another Beta distribution as the normalize term. BNPO can adaptively adjust its parameters in normalization term to match with the evolution of distribution of the expected reward. We theoretically prove that BNPO can effectively reduce the variance in estimating policy gradient. We also that BNPO can reduces to REINFORCE with baseline and GRPO under binary-valued reward circumstance. In order to account for more complex reward systems, we further propose a advantage decomposition mechanism to make BNPO more applicable. Finally, we conduct extensive experiments to verify the effectiveness of our BNPO.

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

## A    FURTHER ELABORATION

**Reasons for Using the Beta Distribution**    We model $p(q)$ using a Beta distribution, as this choice is particularly suitable and widely adopted for modeling probabilities, as discussed in Section 2.1. Moreover, $p(q)$ can be interpreted as the probability that question $q$ is correctly answered.

Additionally, we observe that the normalization terms in REINFORCE and GRPO correspond to Beta distributions. This observation suggests that BNPO, which generalizes existing policy optimization methods, benefits from modeling $p(q)$ with a Beta distribution to achieve improved performance.

While other parameterized distributions could, in principle, be used to model $p(q)$, the Beta distribution offers several distinct advantages:

- **Natural support on** $[0, 1]$**:** The Beta distribution is inherently defined on the interval $[0, 1]$, making it ideal for modeling probabilities. In contrast, many alternative distributions do not possess this property.
- **Ease of parameter estimation:** The parameters of the Beta distribution can be efficiently estimated using Eq.(13). For many other distributions, parameter estimation may be analytically intractable or computationally intensive.
- **Analytical gradient variance minimization:** When $p(q)$ follows a Beta distribution, it is possible to derive an analytical solution for minimizing gradient variance, as shown in Theorem 1. For most other distributions, obtaining such a solution is unlikely.

These reasons collectively motivate our use of the Beta distribution for modeling $p(q)$.

## B    FURTHER EXPERIEMTNS

Table 2: Standard deviation of 3 training runs.

| Methods | MATH500 | AMC23 | AIME2024 | AIME2025 | Avg |
|---|---|---|---|---|---|
| REINFORCE | 0.008 | 0.00392 | 0.003125 | 0.00208 | 0.00031 |
| ReMax | 0.009 | 0.003125 | 0.004165 | **0.0** | 0.000425 |
| GRPO | 0.009 | 0.00625 | 0.00625 | 0.00417 | 0.000165 |
| REINFORCE++ | 0.002 | **0.002555** | 0.00729 | 0.0031265 | 0.00113 |
| BNPO | **0.002** | 0.00781 | **0.003125** | 0.00729 | **0.000155** |

*Note:* Avg is computed by first averaging the performance across the four datasets for each run and then calculating the standard deviation of these averages.

Table 3: Gradient variance.

| Step | 100 | 200 | 300 | 400 | 500 | 600 | 700 | 800 | 900 | 1000 | 1100 |
|---|---|---|---|---|---|---|---|---|---|---|---|
| REINFORCE | 354 | **233** | 127 | 143 | 162 | 118 | 133 | 140 | 196 | 180 | 256 |
| REMAX | 702 | 475 | 385 | 269 | 250.743 | 108 | 130 | 172 | 200 | 404 | 423 |
| GRPO | 2652 | 1552 | 1183 | 505 | 552 | 673 | 537 | 610 | 807 | 606 | 862 |
| REINFORCE++ | 764 | 1455 | 1407 | 612 | 741 | 722 | 1402 | 739 | 1060 | 936 | 1146 |
| BNPO | **235** | 243 | **71** | **123** | **133** | **71** | **115** | **102** | **156** | **131** | **163** |

**Standard deviation**    We present the standard deviations of the performance across different training runs for various methods. Table 2 shows that the standard deviations are low, especially for the standard deviation of average performance (Avg), and BNPO achieves the lowest standard deviations among all methods.

**Gradient variance**    Computing the gradient variance requires sampling multiple batches, which is computationally expensive. Therefore, we compute it only once every 100 training steps. The results, presented in Table 3, show that our BNPO method achieves significantly lower gradient variance compared to the other methods.

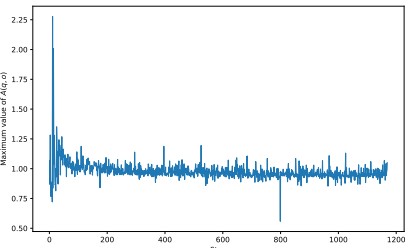

Figure 4: The maximum value of $A_{\alpha,\beta}(q,o)$.

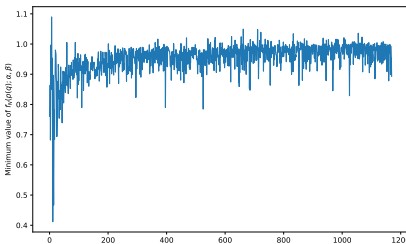

Figure 5: The minimum value of $f_N(p;\alpha,\beta)$.

Table 4: The performance of policy optimization methods with and without advantage decomposition.

| Methods | MATH500 | AMC23 | AIME2024 | AIME2025 | Average |
|---|---|---|---|---|---|
| *Qwen2.5-1.5B-Instruct* | | | | | |
| Base | 14.2 | 7.3 | 1.3 | 0.2 | 5.8 |
| GRPO | 60.0 | 35.5 | 4.6 | 0.8 | 25.2 |
| REINFORCE++ | 58.2 | 32.2 | **6.0** | 1.9 | 24.6 |
| AD-GRPO | **61.6** | 34.1 | 3.8 | **2.9** | 25.6 |
| AD-REINFORCE++ | 58.0 | 35.6 | 4.2 | 1.9 | 24.9 |
| AD-BNPO | 61.4 | **36.3** | 3.8 | 1.9 | **25.8** |

**Advantage decomposition**  To evaluate the effectiveness of the advantage decomposition method, we incorporate an additional format reward following DeepSeek-R1. Since Qwen2.5-Math-1.5B and Qwen2.5-Math-7B exhibit limited instruction-following capabilities, making it difficult for them to learn from the format reward, we use Qwen2.5-1.5B-Instruct as the base model. We denote GRPO and REINFORCE++ with advantage decomposition as AD-GRPO and AD-REINFORCE++, respectively. Note that REINFORCE and ReMax do not include normalization and are therefore excluded from this comparison.

As shown in Table 4, both AD-GRPO and AD-REINFORCE++ achieve slight improvements over their original counterparts. BNPO continues to deliver the best average performance. However, since the format reward surpasses 90% after only 100 training iterations, the overall performance gains from advantage decomposition are relatively modest.

**Advantage function**  Due to the boundness of the reward ($R(q,o) \in \{0,1\}$), normalizing the reward would not lead to instability. To demonstrate it, we show the maximum value of the absolute value of the advantage (or the normalized reward) at each step. As shown in Figure 4, the normalized reward always remains at a relatively small value. We also show the minimum value of the normalization term $f_N(p;\alpha,\beta)$ at each step. As shown in Figure 5, the minimum value of $f_N(p;\alpha,\beta)$ always remained consistently around 1.

**Performance**  Since the best performance for different datasets may be achieved at different training steps. The reported best average performance does not mean that it is the best performance of each dataset. Thus, we further report the best performance of each individual dataset during training. As shown in the Table 5, BNPO can achieves consistent improvement on different datasets and achieve good performance on complex datasets.

**Beta distribution**  Since we model $p(q)$ as a random variable with Beta distribution $f_D(p(q);a,b)$, we evaluate how well this distribution fits the empirical data. To evaluate estimation error, we use the average of log-likelihood to measure the goodness of fit. As shown in Figure 6, the statistic remains within a relatively small range in most steps, indicating that the fitted distribution matches the empirical distribution and introduces a small amount of bias.

Table 5: The performance of different policy optimization methods on math datasets.

| Methods | MATH500 | AMC23 | AIME2024 | AIME2025 |
|---|---|---|---|---|
| *Qwen2.5-Math-1.5B* | | | | |
| Base | 28.0 | 27.3 | 6.0 | 3.1 |
| REINFORCE | 74.8 | 54.9 | 18.3 | 11.9 |
| ReMax | 74.4 | 54.7 | 19.0 | 11.0 |
| GRPO | 74.6 | 54.8 | 19.0 | 11.0 |
| REINFORCE++ | **75.4** | 53.8 | 19.2 | 10.4 |
| BNPO | 75.0 | **55.0** | **19.8** | **13.1** |
| *Qwen2.5-Math-7B* | | | | |
| Base | 41.4 | 32.5 | 11.0 | 5.0 |
| REINFORCE | 79.6 | 66.1 | 33.1 | 15.0 |
| ReMax | 79.6 | 66.6 | **33.5** | 15.4 |
| GRPO | **81.6** | 65.7 | 32.5 | 13.3 |
| REINFORCE++ | 79.8 | 65.0 | 32.3 | 13.1 |
| BNPO | 79.8 | **68.8** | **33.5** | **15.6** |

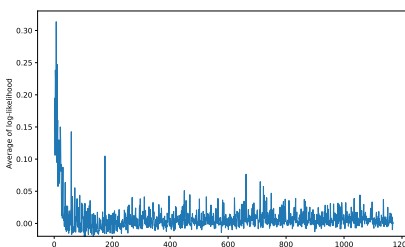

Figure 6: The average of log-likelihood.

**The values of** $(\mathbb{E}[p(q)], \mathrm{Var}[p(q)])$   We show the curves of $E[p]$ and $\mathrm{Var}[p]$ in Figure 7 and Figure 8 separately, which clearly show that both quantities vary throughout training. Beta Normalization is introduced precisely to adapt to this dynamic behavior and to mitigate the resulting instability in the distribution of $p$.

## C  PROOF

**Theorem 1.** *Let $q \sim \rho$ be a question and $o \sim \pi_\theta(\cdot|q)$ be an output with reward $R(q, o) \in \{0, 1\}$, where $R(q, o)$ follows a Bernoulli distribution with success probability $p(q) = \mathbb{E}_{o \sim \pi_\theta(\cdot|q)}[R(q, o)|q]$, and that $p(q)$ follows a Beta distribution $f_D(p(q); a, b)$. Define the BNPO gradient estimator as*

$$g_{\alpha,\beta} = \nabla_\theta \log \pi(o|q) \frac{R(q, o) - p(q)}{f_N(p(q); \alpha, \beta)}.$$

*where $f_N(p(q); \alpha, \beta)$ is a Beta distribution. Under the assumption $\nabla_\theta \log \pi(o|q)$ is uncorrelated with $\frac{R(q,o)-p(q)}{f_N(p(q);\alpha,\beta)}$, the variance of the policy gradient estimator $\mathrm{Var}_{q \sim \rho, \, o \sim \pi_\theta(\cdot|q)}(g_{\alpha,\beta})$ is finite if and only if: $\alpha < \frac{a+3}{2}$ and $\beta < \frac{b+3}{2}$. Within this domain, $\mathrm{Var}_{q \sim \rho, \, o \sim \pi_\theta(\cdot|q)}(g_{\alpha,\beta})$ attains a unique minimum at:*

$$\alpha = 1 + \frac{a}{3}, \quad \beta = 1 + \frac{b}{3}.$$

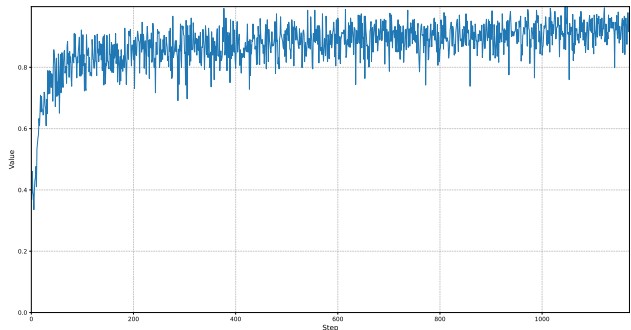

Figure 7: The values of $\mathbb{E}[p(q)]$ during training.

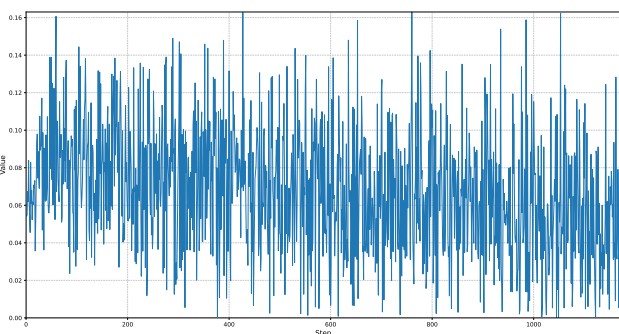

Figure 8: The values of $\mathrm{Var}[p(q)]$ during training.

*Proof.* 1. VARIANCE EXPRESSION

Expand the variance using its definition:

$$\mathrm{Var}(g_{\alpha,\beta}) = \mathbb{E}\left[\left(\nabla_\theta \log \pi(o \mid q) \cdot \frac{R(q,o) - p(q)}{f_N(p(q);\alpha,\beta)}\right)^2\right]$$

$$- \left(\mathbb{E}\left[\nabla_\theta \log \pi(o \mid q) \cdot \frac{R(q,o) - p(q)}{f_N(p(q);\alpha,\beta)}\right]\right)^2.$$

Simplify the mean term using the assumption and $\mathbb{E}[\frac{R(q,o)-p(q)}{f_N(p(q);\alpha,\beta)}|q] = 0$:

$$\mathbb{E}\left[\nabla_\theta \log \pi(o \mid q) \cdot \frac{R(q,o) - p(q)}{f_N(p(q);\alpha,\beta)}\right] = 0.$$

Therefore:

$$\mathrm{Var}(g_{\alpha,\beta}) = \mathbb{E}\left[(\nabla_\theta \log \pi(o \mid q))^2 \cdot \frac{(R(q,o) - p(q))^2}{f_N(p(q);\alpha,\beta)^2}\right].$$

Under the assumption, the variance of the gradient estimator $g_{\alpha,\beta}$ is proportional to:

$$\mathrm{Var}(g_{\alpha,\beta}) \propto \mathbb{E}_{q\sim\rho}\mathbb{E}_o\left[\frac{(R(q,o) - p(q))^2}{f_N(p(q);\alpha,\beta)^2}\right].$$

For $R(q,o) \in \{0,1\}$, we have that

$$\mathbb{E}_o\left[(R - p(q))^2|q\right] = p(q)(1 - p(q)).$$

Substituting the weight function:

$$\frac{p(q)(1-p(q))}{f_N(p(q);\alpha,\beta)^2} = \frac{p(q)(1-p(q))}{\left(\frac{1}{B(\alpha,\beta)}p(q)^{\alpha-1}(1-p(q))^{\beta-1}\right)^2}$$

$$= B(\alpha,\beta)^2 \cdot p(q)^{3-2\alpha}(1-p(q))^{3-2\beta}.$$

Unper $p \sim f_D(p(q); a, b)$, the expectation integrates the above expression over the Beta-distributed $p$ becomes

$$\mathbb{E}_p\left[p^{3-2\alpha}(1-p)^{3-2\beta}\right] = \frac{B(a+3-2\alpha, b+3-2\beta)}{B(a,b)},$$

Thus, we have that

$$\mathrm{Var}(g_{\alpha,\beta}) \propto B(\alpha,\beta)^2 \cdot \frac{B(a+3-2\alpha, b+3-2\beta)}{B(a,b)}.$$

### 2. DOMAIN OF FINITENESS

The Beta function $B(x, y)$ converges iff $x > 0$ and $y > 0$. For convergence of $B(a + 3 - 2\alpha, b + 3 - 2\beta)$:

$$a + 3 - 2\alpha > 0 \implies \alpha < \frac{a+3}{2},$$

$$b + 3 - 2\beta > 0 \implies \beta < \frac{b+3}{2}.$$

### 3. BOUNDARY BEHAVIOR

As $\alpha \to \frac{a+3}{2}^-$ or $\beta \to \frac{b+3}{2}^-$:

$$B(a + 3 - 2\alpha, b + 3 - 2\beta) \to \infty \implies \mathrm{Var}(g_{\alpha,\beta}) \to +\infty.$$

### 4. OPTIMAL PARAMETERS

Define $L(\alpha, \beta) = \ln \mathrm{Var}(g_{\alpha,\beta})$:

$$L = 2\ln B(\alpha,\beta) + \ln B(a + 3 - 2\alpha, b + 3 - 2\beta) - \ln B(a,b).$$

The partial derivatives are:

$$\frac{\partial L}{\partial \alpha} = 2\left[\psi(\alpha) - \psi(\alpha+\beta)\right] - 2\left[\psi(a+3-2\alpha) - \psi(a+b+6-2\alpha-2\beta)\right],$$

$$\frac{\partial L}{\partial \beta} = 2\left[\psi(\beta) - \psi(\alpha+\beta)\right] - 2\left[\psi(b+3-2\beta) - \psi(a+b+6-2\alpha-2\beta)\right],$$

where $\psi(x) = \frac{d}{dx}\ln\Gamma(x)$ and $\Gamma(x)$ is the gamma function. Setting $\partial L/\partial\alpha = 0$ and $\partial L/\partial\beta = 0$:

$$\psi(\alpha) - \psi(\alpha+\beta) = \psi(a+3-2\alpha) - \psi(a+b+6-2\alpha-2\beta).$$

$$\psi(\beta) - \psi(\alpha+\beta) = \psi(b+3-2\beta) - \psi(a+b+6-2\alpha-2\beta).$$

Substituting $\alpha = 1 + \frac{a}{3}$ and $\beta = 1 + \frac{b}{3}$ satisfies this identity through digamma function properties.

### 5. STRICT CONVEXITY

We compute the Hessian matrix $H$ for $L(\alpha, \beta)$ at $(\alpha_0, \beta_0)$.

Let $\psi_1(x) = \frac{d}{dx}\psi(x)$ be the trigamma function. Let $X_\alpha = \alpha_0 = 1 + a/3$ and $X_\beta = \beta_0 = 1 + b/3$. Let $S_{sum} = \alpha_0 + \beta_0 = 2 + (a+b)/3$.

The arguments for the other digamma terms at the solution become: $a - 2\alpha_0 + 3 = 1 + a/3 = X_\alpha$. $b - 2\beta_0 + 3 = 1 + b/3 = X_\beta$. $a + b - 2\alpha_0 - 2\beta_0 + 6 = (a+b)/3 + 2 = S_{sum}$.

The second partial derivatives are:

$$\frac{\partial^2 L}{\partial \alpha^2} = 2\psi_1(\alpha) - 2\psi_1(\alpha + \beta) + 4\psi_1(a - 2\alpha + 3) - 4\psi_1(a + b - 2\alpha - 2\beta + 6).$$

At $(\alpha_0, \beta_0)$: $H_{11} = 2\psi_1(X_\alpha) - 2\psi_1(S_{sum}) + 4\psi_1(X_\alpha) - 4\psi_1(S_{sum}) = 6\psi_1(X_\alpha) - 6\psi_1(S_{sum})$. By symmetry: $H_{22} = 6\psi_1(X_\beta) - 6\psi_1(S_{sum})$. The mixed partial derivative:

$$\frac{\partial^2 L}{\partial \alpha \partial \beta} = -2\psi_1(\alpha + \beta) - (-2)\psi_1(a + b - 2\alpha - 2\beta + 6)(-2)$$

$$= -2\psi_1(\alpha + \beta) - 4\psi_1(a + b - 2\alpha - 2\beta + 6).$$

At $(\alpha_0, \beta_0)$: $H_{12} = -2\psi_1(S_{sum}) - 4\psi_1(S_{sum}) = -6\psi_1(S_{sum})$.

For a minimum, $H$ must be positive definite.

1. $H_{11} > 0$: $H_{11} = 6(\psi_1(X_\alpha) - \psi_1(S_{sum})) = 6(\psi_1(1 + a/3) - \psi_1(2 + (a + b)/3))$. Since $a, b > 0$, we have $X_\beta = 1 + b/3 > 0$. Thus $X_\alpha = 1 + a/3 < 1 + a/3 + (1 + b/3) = S_{sum}$. The trigamma function $\psi_1(x)$ is strictly decreasing for $x > 0$. Since $X_\alpha < S_{sum}$ (and $X_\alpha, S_{sum} > 0$), $\psi_1(X_\alpha) > \psi_1(S_{sum})$. Thus $H_{11} > 0$. Similarly $H_{22} > 0$.

2. $\det(H) = H_{11}H_{22} - H_{12}^2 > 0$:

$\det(H)$

$= (6\psi_1(X_\alpha) - 6\psi_1(S_{sum}))(6\psi_1(X_\beta) - 6\psi_1(S_{sum})) - (-6\psi_1(S_{sum}))^2$

$= 36[\psi_1(X_\alpha)\psi_1(X_\beta) - \psi_1(X_\alpha)\psi_1(S_{sum}) - \psi_1(X_\beta)\psi_1(S_{sum}) + \psi_1(S_{sum})^2 - \psi_1(S_{sum})^2]$

$= 36[\psi_1(X_\alpha)\psi_1(X_\beta) - \psi_1(S_{sum})(\psi_1(X_\alpha) + \psi_1(X_\beta))].$

For $\det(H) > 0$, we need $\psi_1(X_\alpha)\psi_1(X_\beta) - \psi_1(S_{sum})(\psi_1(X_\alpha) + \psi_1(X_\beta)) > 0$. Since $\psi_1(x) > 0$ for $x > 0$, we can divide by $\psi_1(X_\alpha)\psi_1(X_\beta)\psi_1(S_{sum})$:

$$\frac{1}{\psi_1(S_{sum})} - \left(\frac{1}{\psi_1(X_\beta)} + \frac{1}{\psi_1(X_\alpha)}\right) > 0 \implies \frac{1}{\psi_1(X_\alpha + X_\beta)} > \frac{1}{\psi_1(X_\alpha)} + \frac{1}{\psi_1(X_\beta)}.$$

Let $f(x) = 1/\psi_1(x)$. The inequality is $f(X_\alpha + X_\beta) > f(X_\alpha) + f(X_\beta)$. The function $f(x) = 1/\psi_1(x)$ is strictly convex on $(0, \infty)$.

As $x \to 0^+$, $\psi_1(x) \to \infty$, so $f(x) = 1/\psi_1(x) \to 0$. So we can define $f(0) = 0$. For a strictly convex function $f$ with $f(0) = 0$: For $x, y > 0$, $f(x) = f(\frac{x}{x+y}(x + y) + \frac{y}{x+y} \cdot 0) < \frac{x}{x+y}f(x + y) + \frac{y}{x+y}f(0) = \frac{x}{x+y}f(x + y)$. Similarly, $f(y) < \frac{y}{x+y}f(x + y)$. Summing these gives $f(x) + f(y) < f(x + y)$. The strict inequality holds because $X_\alpha = 1 + a/3 > 0$ and $X_\beta = 1 + b/3 > 0$. Therefore, the Hessian matrix is positive definite at $(\alpha_0, \beta_0)$. Since the domain for $(\alpha, \beta)$ (where variance is finite, and $\alpha, \beta > 0$) is a convex set, this implies that $(\alpha_0, \beta_0)$ is a unique minimum. $\square$

