# OpenReview forum: "BNPO: Beta Normalization Policy Optimization"
_ICLR.cc/2026/Conference — ICLR 2026 Conference Withdrawn Submission_

### Official Review · Reviewer_qMA3 · 2025-10-16

**Soundness:** 2
**Presentation:** 4
**Contribution:** 3
**Rating:** 6
**Confidence:** 4

**Summary:**

The authors propose a dynamic, beta-distribution based normalization technique for the Advantage function that is used in a policy gradient context. Their method is specifically designed for binary reward settings. The authors call this method BNPO and use it train LLMs which are then evaluated on several math benchmarks.

**Strengths:**

- The empirical results show promise, however, as mentioned in the weaknesses they need to be presented more precisely
- The authors give an extensive background section, making even a reader that is unfamiliar with the topic able to follow the contents of the paper. However, in case you might need extra space to properly include the reviewers feedack, I think you could prune the beta-distribution section or move it to the appendix. I think that that is more or less basic math knowledge and anyone unfamiliar with it, could quickly read it up in a math textbook.
- I quite like the general idea and it is quite elegant that one is able to unifying REINFORCE and GRPO in one framework.

**Weaknesses:**

- Insufficient related work: Briefly touch in a broader scope of the application of normalization in RL. E.g. observation, reward normalization (i.e. before being processed by the training algorithm) and for example SAC (Soft Actor-Critic Algorithms and Applications, Haarnoja) dynamically normalizes its entropy coefficient.
- Missing Confidence Intervals for Empirical Results
- I am not convinced that the average in Table 1 is sufficient to say that BNPO is best overall. At the end of the day, BNPO is only best only in AMC23 and the difficulty of squeezing out an improvement differs from dataset to dataset. For example, the 0.2 gain of REINFORCE over BNPO in AIME2025 is percentagewise as significant as the lead GRPO has over BNPO in MATH500, however it almost does not contribute to the average value. For transparency I think one would also have to include different metrics aside from the average such as avg % increase.

**Questions:**

- Please be more clear about your hyperparamter choice in "Metrics" you say that they are based on the best average performance? Of which method? Then later you say that for fairness all methods use the same hyperparamter which would not be fair if they are tuned for one specific method
- Could you include runtime comparisons? BNPO uses Monte-Carlo samples to compute its parameters alpha and beta which might hamper the runtime. Please elaborate.
- While you give a good intution that p(q) follows a beta-distribution, to substantiate this claim, it would be good to actually plot the empirical distribution of some ps and show that it can be well-fitted with some beta-distribution
- I do not get Lines 256-259. To reduce to RLOO, F_N would have to be parametrisable to be any constant, however the only constant F_N can be is 1?
- Are there any empirical validations for the claim in Lines 297
- Training stability: Isnt it possible to just directly compute the policy variance instead of using a proxy?

Minors:
- Missing space in Line 134 before sigma and after 'end'
- Missing formal problem description. I think a multi-armed bandit would be the problem formalization here. At least mention it.

---

> ### Author Response · Authors · 2025-11-25
>
> We appreciate your careful and constructive comments. We have addressed the questions that you raised as follows. Please let us know if you have any further concerns.
>
> $\textbf{Q1:}$ Insufficient related work: Briefly touch in a broader scope of the application of normalization in RL. E.g. observation, reward normalization (i.e. before being processed by the training algorithm) and for example SAC (Soft Actor-Critic Algorithms and Applications, Haarnoja) dynamically normalizes its entropy coefficient.
>
> $\textbf{A1:}$ Due to space constraints, we primarily focused on works most relevant to our study. In the revision, we have expanded the Related Work section to provide a more comprehensive discussion.
>
> $\textbf{Q2:}$ Missing Confidence Intervals for Empirical Results.
>
> $\textbf{A2:}$ We did not omit the confidence intervals. Due to space constraints, they are provided in the appendix (Page 12, Table 2). We report the standard deviations of performance across different training runs. As shown in Table 2, the standard deviations are generally low, particularly for the average performance (Avg), and BNPO achieves the lowest standard deviations among all methods.
>
> $\textbf{Q3:}$ I am not convinced that the average in Table 1 is sufficient to say that BNPO is best overall. At the end of the day, BNPO is only best only in AMC23 and the difficulty of squeezing out an improvement differs from dataset to dataset. For example, the 0.2 gain of REINFORCE over BNPO in AIME2025 is percentagewise as significant as the lead GRPO has over BNPO in MATH500, however it almost does not contribute to the average value. For transparency I think one would also have to include different metrics aside from the average such as avg % increase.
>
> $\textbf{A3:}$ Since the optimal performance for different datasets may occur at different training steps, the reported best average performance does not necessarily reflect the best performance on each individual dataset. To address this, we additionally report the best performance achieved on each dataset during training. As shown in Table 5 (Page 14), BNPO consistently improves performance across datasets and achieves good results even on complex datasets.
>
> $\textbf{Q4:}$ Please be more clear about your hyperparamter choice in "Metrics" you say that they are based on the best average performance? Of which method? Then later you say that for fairness all methods use the same hyperparamter which would not be fair if they are tuned for one specific method.
>
> $\textbf{A4:}$ We have clarified the description to avoid misunderstandings. We do not tune hyperparameters for specific methods; the hyperparameter settings generally follow common defaults.
>
> There are three hyperparameters related to training: batch size, learning rate, and the number of rollouts per question. We set the batch size to 32, the learning rate to $10^{-6}$, and the number of rollouts to 16, following DeepSeek-R1. Two hyperparameters are related to evaluation: we set the temperature to 0.6 and top-p to 0.95, following the recommended settings for Qwen series models.
>
> We have also provided the source code. To run experiments with different policy optimization methods, simply modify line 52 in the file recipe/bnpo/run.sh by setting adv_estimator="bnpo".
>
> The early stopping step is determined based on average performance. We evaluate performance every 16 training steps and report the step with the best average performance during training.
>
> $\textbf{Q5:}$ Could you include runtime comparisons? BNPO uses Monte-Carlo samples to compute its parameters alpha and beta which might hamper the runtime. Please elaborate.
>
> $\textbf{A5:}$ The runtimes of different policy optimization methods are shown in the table below. ReMax requires approximately 25% more runtime than the other methods. As shown in Table 1 and Figure 2, BNPO achieves the best performance and stability while maintaining comparable training efficiency.
> | Method       | Time  |
> |--------------|--------|
> | REINFORCE    | 23.9h  |
> | REINFORCE++  | 23.9h  |
> | GRPO         | 24.2h  |
> | ReMax        | 30.4h  |
> | BNPO         | 23.9h  |

---

> ### Author Response · Authors · 2025-11-25
>
> $\textbf{Q6:}$ While you give a good intution that p(q) follows a beta-distribution, to substantiate this claim, it would be good to actually plot the empirical distribution of some ps and show that it can be well-fitted with some beta-distribution.
>
> $\textbf{A6:}$ Since we model $p(q)$ as a random variable with Beta distribution $f_{D}(p(q);a,b)$, we evaluate how well this distribution fits the empirical data. To evaluate estimation error, we use the average of log-likelihood to measure the goodness of fit. As shown in Figure 6 (Page 14), the statistic remains within a relatively small range in most steps, indicating that the fitted distribution matches the empirical distribution and introduces a small amount of bias.
>
> $\textbf{Q7:}$ I do not get Lines 256-259. To reduce to RLOO, F_N would have to be parametrisable to be any constant, however the only constant F_N can be is 1?
>
> $\textbf{A7:}$ Setting $\alpha$ and $\beta$ to 1 reduces BNPO to RLOO. If $\alpha \neq 1$ or $\beta \neq 1$, the value of $f_{N}(p(q); \alpha, \beta)$ remains influenced by $p(q)$, whereas in RLOO, the normalization term is always 1 (i.e., no normalization), resulting in a fundamental difference.
>
> $\textbf{Q8:}$ Are there any empirical validations for the claim in Lines 297.
>
> $\textbf{A8:}$ 	We do not aim to conduct experiments with binary-valued rewards, as DeepSeek-R1 has already demonstrated that such rewards can yield strong RL performance. To evaluate the effectiveness of BNPO under continuous or multi-valued rewards, we introduce an additional length reward [1]. The results, shown in the table below, indicate that BNPO remains effective in this setting.
>
> | Method       | Acc|Length|
> |--------------|--------|--------|
> | REINFORCE    |38.3  | 785  |
> | REINFORCE++  | 38.2  | 784  |
> | GRPO         | 38.3  | 744  |
> | ReMax        | 37.5 | 795  |
> | BNPO         | 38.5  |784  |
>
> [1] Yu Q, Zhang Z, Zhu R, et al. Dapo: An open-source llm reinforcement learning system at scale[J]. arXiv preprint arXiv:2503.14476, 2025.
>
> $\textbf{Q9:}$ Training stability: Isn’t it possible to just directly compute the policy variance instead of using a proxy?
>
> $\textbf{A9:}$ We have shown the gradient variance directly. Due to space constraint, we put it in the appendix (Page 12 Table 3). Computing the gradient variance requires sampling multiple batches, which is computationally expensive. Therefore, we compute it only once every 100 training steps. The results, presented in Table 3, show that our BNPO method achieves significantly lower gradient variance compared to the other methods.

---

> > ### Comment · Reviewer_qMA3 · 2025-11-25
> > **Answer**
> >
> > Dear authors,
> >
> > thank you very much for addressing these unclarities which I believe will benefit the quality of the final paper. Nontheless, I do not believe that these changes would justify a rating change which is why I stick to my current rating.

---

### Official Review · Reviewer_9sLr · 2025-10-22

**Soundness:** 1
**Presentation:** 2
**Contribution:** 1
**Rating:** 2
**Confidence:** 4

**Summary:**

This paper examines reward normalization methods for REINFORCE-type policy gradient algorithms, primarily in the context of binary reward problems. The paper proposes  Beta normalization, which normalizes the center reward $R - p(q)$ by a Beta distribution PDF with adapted parameters, where $p(q) = E[R|q]$ is the success probability for question $q$. Some conceptual justification is given, and experiments using small language models on MATH are conducted to show the performance of the new normalization method.

**Strengths:**

1. The paper proposes a complete workflow for normalizing the reward for policy gradient with a Beta distribution, including estimating the distribution parameters.
2. It is interesting that with a different parameter setup, Beta normalization recovers REINFORCE and GRPO.
3. The proposed approach is empirically evaluated on multiple datasets and multiple models, with both pass rate and gradient norm.

**Weaknesses:**

The conceptual aspect of the proposed method, in many aspects, is somewhat flawed, which limits the method in general RL problems.

1. The correctness of the gradient estimator is questionable. After the new normalization, the gradient estimator is no longer unbiased to the exact policy gradient. There is no conceptual justification that the new estimator would achieve a solution close to the optimal policy of the original problem for general problems.

2. The theorem in this paper rests on an unrealistic (even erroneous) independence assumption and is vacuous. The core variance-reduction justification assumes $\nabla_\theta \log \pi(o|q)$ is uncorrelated with $\frac{R(q,o) - p(q)}{f_N(p(q); \alpha, \beta)}$. This is generally false in policy gradient because both terms have explicit dependence on $\pi_\theta$. Therefore, the proof does not convincingly justify the claimed minimum variance and stability.

3. The modeling of success probability $p(q)$ over the population of questions with a single Beta distribution is fragile, especially when the quantity is non-stationary and heavily depends on the policy $\pi_\theta$. There is no justification, both theoretically and empirically, for how the modeling is correct, and what would happen if the modeling fails. It is largely unclear why using a Beta distribution density function to normalize would result in a better performance, and under what circumstances it will and will not.

4. Normalizing the reward by a density function could lead to instability. The estimated $p(q)$ may correspond to a small density, which amplifies the noise in advantage estimation. Therefore, the gap between this possibility and the reported stability in empirical evaluation is unclear.

The empirical evaluation is also limited to demonstrating the performance of the proposed method in general applications.

1. The empirical evaluation misses the core evaluation of the value in the proposed Beta normalization. The authors did not demonstrate how the empirical $p(q)$ fits the beta distribution across training periods, or how the curves of gradient variance vary across time compared to baselines, or how sensitive the performance is to mis-specified $(a,b)$ in estimation, especially when the sampled $p(q)$ gives a small density.

2. The reported score of the proposed method is only marginally higher than some benchmarks, and not the best in several individual benchmarks. Moreover, with Beta normalization, the method still performs quite poorly in slightly more complex tasks such as AIME. It is unclear whether the normalization benefit (if there is any) could scale to more complex IMO-style tasks, or modern large models with many more parameters, or even to more diverse reasoning tasks, such as coding and science reasoning. The marginality and inconsistency do not translate to a convincing motivation for using this method in reality for general RL problems.

**Questions:**

see weaknesses

---

> ### Author Response · Authors · 2025-11-25
>
> We appreciate your careful and constructive comments. We have addressed the questions that you raised as follows. Please let us know if you have any further concerns.
>
> $\textbf{Q1:}$ After the new normalization, the gradient estimator is no longer unbiased to the exact policy gradient. There is no conceptual justification that the new estimator would achieve a solution close to the optimal policy of the original problem for general problems.
>
> $\textbf{A1:}$ Although BNPO may introduce a small amount of bias, this does not affect convergence in theory. Since the normalization term is identical for all outputs $o$ corresponding to the same question, the normalization term only varies across different questions and does not alter the relative preference among outputs within a question. Moreover, the normalization term is always positive, ensuring that it does not change the optimal solution; it merely adjusts the learning priority among questions. Therefore, BNPO preserves the convergence behavior. The empirical results in Table 1 also verify this.
>
> $\textbf{Q2:}$ The theorem in this paper rests on an unrealistic independence assumption and is vacuous.
>
> $\textbf{A2:}$ Since the reward function is typically defined independently of the policy network, the correlation between the reward signal and the policy is generally weak. For instance, the reward function defined in Eq.(9) depends solely on the reference answer and the model output, without involving the policy parameters.
>
> This assumption is also commonly adopted in reinforcement learning theory. A notable example is the derivation of the optimal state-dependent baseline, expressed in Eq.(12) in [1]. Although theoretically optimal, this baseline is seldom used in practice due to its complexity. Instead, it is common to assume that $\nabla_\theta \log \pi(o | q)$ is uncorrelated with $Q(s,a)$, which leads to the widely used baseline $b(s_t)=V(s_t)$ (Pleaser refer to Section 4.2 and Section 4.4 of [1]).
>
> The independence assumption is introduced primarily for analytical tractability as in [1]. However, even without this assumption, the result in Theorem 1 remains valid as a suboptimal but effective strategy for variance reduction.
>
> This is analogous to REINFORCE with a state-dependent baseline: although the baseline $b(s_t)=V(s_t$) may not be the optimal state-dependent baseline in the absence of independence, it can still reduce gradient variance.
>
> We also verify the effectiveness of reducing gradient variance by experiments. As shown in the Table, BNPO achieves the lowest gradient variance than alternative methods, which aligns with our theoretical analysis.
>
> [1] Wu C, Rajeswaran A, Duan Y, et al. Variance reduction for policy gradient with action-dependent factorized baselines[J]. arXiv preprint arXiv:1803.07246, 2018.
>
> $\textbf{Q3:}$ Normalizing the reward by a density function could lead to instability.
>
> $\textbf{A3:}$ Due to the boundness of the reward ($R(q,o)\in \{0,1\}$), normalizing the reward would not lead to instability. To demonstrate it, we show the maximum value of the absolute value of the advantage (or the normalized reward) at each step. As shown in Figure 4 (Page 13), the normalized reward always remains at a relatively small value. We also show the minimum value of the normalization term $f_{N}(p;\alpha,\beta)$ at each step. As shown in Figure 5 (Page 13), the minimum value of $f_{N}(p;\alpha,\beta)$ always remained consistently around 1.

---

> ### Author Response · Authors · 2025-11-25
>
> $\textbf{Q4:}$ The authors did not demonstrate how the empirical fits the beta distribution across training periods, or how the curves of gradient variance vary across time compared to baselines, or how sensitive the performance is to mis-specified in estimation, especially when the sampled gives a small density.
>
> $\textbf{A4:}$ To evaluate estimation error, we use the average of log-likelihood to measure the goodness of fit. As shown in Figure 6 (Page 14), the value remains within a relatively small range in most steps, indicating that the fitted distribution matches the empirical distribution and introduces a small amount of bias.
>
> We have shown the gradient variance. Due to space constraint, we put it in the appendix (Page 12 Table 3). The results, presented in Table 3, show that our BNPO method achieves significantly lower gradient variance compared to the other methods.
>
> For the effect of the density function in BNPO, please refer to our response A2.
>
> $\textbf{Q5:}$ The reported score of the proposed method is only marginally higher than some benchmarks, and not the best in several individual benchmarks. Moreover, with Beta normalization, the method still performs quite poorly in slightly more complex tasks such as AIME.
>
> $\textbf{A5:}$ Since the optimal performance for different datasets may occur at different training steps, the reported best average performance does not necessarily reflect the best performance on each individual dataset. To address this, we additionally report the best performance achieved on each dataset during training. As shown in Table 5 (Page 14), BNPO consistently improves performance across datasets and achieves good results even on complex datasets.
>
> Beyond performance improvements, BNPO provides noticeably more stable training compared to other methods, which is important when training large language models. As shown in Figure 2, BNPO maintains a consistently stable gradient norm throughout training, with minimal fluctuation.

---

> > ### Comment · Reviewer_9sLr · 2025-11-27
> >
> > I thank the authors for their responses and additional experiment details. However, I don't find the authors' response convincing enough to resolve my concerns, and I feel there are some overclaims and shaky arguments in the responses as well.
> >
> > For the correctness, I believe the paper does not provide any theoretical convergence results, and thus I don't quite understand how the authors claim "Although BNPO may introduce a small amount of bias, this does not affect convergence in theory," or "BNPO preserves the convergence behavior." With the normalization, the landscape of the objective function is largely changed. I don't see why the method would still converge, even to a stationary point of the original problem, let alone the same global optimal point, which requires formal proof. The distribution of responses depends on the policy, and both $\nabla \log \pi_\theta(o|q)$ and $R(q,o) - p(q) / f_N(p(q), \alpha, \beta)$ depend on the samples from $\pi_\theta$. The reward does not depend on $\theta$ does not necessarily mean the correlation is weak. It is also not common in RL theory to make fundamentally incorrect assumptions when deriving a rigorous theorem. Even though the reward is bounded, the inverse of a density could still blow up. The responses look like a strong and problematic overclaim to me.
> >
> > In general, the paper and the responses together seem to show a lack of fundamental rigor, with components on the edge of correct and incorrect. I'm not confident that the proposed method could fundamentally help the community, and more importantly, the marginal improvement in experiments makes it difficult for me to view this paper as a pure empirical one and ignore these weaknesses.

---

> > > ### Author Response · Authors · 2025-11-28
> > >
> > > Thanks for your responses. However, some of your comments appear to be based on absolute skepticism rather than concrete evidence, which is unfair to the authors.
> > >
> > > $\textbf{Q1:}$ "BNPO preserves the convergence behavior."
> > >
> > > $\textbf{A1:}$ Thanks for your comments. We acknowledge that our previous response contained some bias. Our main goal is to demonstrate that BNPO can still converge, rather than to claim that BNPO achieves the same optimal solution as REINFORCE.
> > >
> > > Let
> > > $\tilde{R}(q,o) = \frac{R(q,o)}{f_{N}(p(q);\alpha,\beta)}$,
> > > and we have
> > > $A(q,o) = \tilde{R}(q,o) - \tilde{p}(q)$,
> > > where
> > > $\tilde{p}(q) = \mathbb{E}\_{o \sim \pi_{\theta}(\cdot|q)}[\tilde{R}(q,o) \mid q]$.
> > > Then, we can treat BNPO as optimizing the new objective
> > > $\mathcal{L}(\theta) = \mathbb{E}\_{q \sim \rho, o \sim \pi\_{\theta}(\cdot|q)}[\tilde{R}(q,o)]$.
> > > Thus, the difference between BNPO and REINFORCE lies in the definition of the reward function. Consequently, BNPO can still converge, which is also validated by the experimental results in Table 1.
> > >
> > > $\textbf{Q2:}$ "It is also not common in RL theory to make fundamentally incorrect assumptions when deriving a rigorous theorem."
> > >
> > > $\textbf{A2:}$ The reward function is typically defined independently of the policy network, which can weaken their correlation. We cannot guarantee that the terms are absolute independent; otherwise, we would state it as a fact rather than an assumption.
> > >
> > > You still miss our provided references to leave a comment. The independence assumption has been employed in [1] and is introduced primarily for analytical tractability. Even without this assumption, the result in Theorem 1 remains valid as a suboptimal yet effective strategy for variance reduction, as also noted in [1]. Our empirical results in Figure 2 and Table 3 further support this conclusion.
> > >
> > > [1] Wu C, Rajeswaran A, Duan Y, et al. Variance reduction for policy gradient with action-dependent factorized baselines[J]. arXiv preprint arXiv:1803.07246, 2018.
> > >
> > > $\textbf{Q3:}$ "Even though the reward is bounded, the inverse of a density could still blow up."
> > >
> > > $\textbf{A3:}$ Due to the boundedness of the reward and the properties of the Beta distribution, the resulting density is always positive (the cases p(q)=0 and p(q)=1 are excluded because they imply R(q,o)-p(q)=0). Figure 5 reports the minimum observed density values, all of which are greater than 0.4, confirming that the density does not approach zero in practice.
> > >
> > > It is unclear what specific threshold would constitute a “blow-up” without a defined criterion. Moreover, our implementation includes a clamping operation to further enhance numerical stability (Please refer to the file "recipe/bnpo/src/bnpo_ray_trainer.py" Line 105 in the supplementary materials), although no clamping was actually needed during the experiments.

---

### Official Review · Reviewer_xn5Y · 2025-10-27

**Soundness:** 3
**Presentation:** 3
**Contribution:** 3
**Rating:** 6
**Confidence:** 2

**Summary:**

The paper introduces Beta Normalization Policy Optimization (BNPO), a novel method for policy optimization in reinforcement learning (RL). The proposed method addresses a critical limitation in current RL policy optimization techniques—namely, their inability to dynamically adjust reward normalization as the policy evolves during training. BNPO utilizes a Beta distribution to adaptively normalize rewards, improving gradient stability and reducing variance in gradient estimates. Theoretical analysis and experimental results demonstrate that BNPO performs better than existing methods, including REINFORCE, GRPO, and REINFORCE++, especially on reasoning tasks.

**Strengths:**

1. BNPO introduces a dynamic reward normalization technique using the Beta distribution, offering a novel solution to reward normalization issues in reinforcement learning. This is particularly important as existing methods use static normalization, which doesn’t adapt as training progresses.
2. The paper provides a solid theoretical analysis showing that BNPO effectively reduces gradient variance and enhances the stability of training. The detailed proof and derivations offer a clear understanding of the method's benefits.
3. The experiments on large-scale language models and reasoning tasks clearly show that BNPO outperforms other state-of-the-art optimization methods, including REINFORCE, GRPO, and REINFORCE++.
4. The advantage decomposition mechanism introduced in the paper allows BNPO to handle more complex reward systems, extending its applicability beyond binary-valued rewards.
5. Experimental results demonstrate that BNPO leads to more stable training dynamics compared to other methods, as evidenced by consistent gradient norms and low gradient variance, which is crucial for large model training.

**Weaknesses:**

1. While BNPO works well for binary-valued rewards, the extension to multi-valued or continuous rewards is mentioned but not thoroughly explored in the paper. More detailed analysis and experiments on this extension would strengthen the claim of BNPO's general applicability.
2. The adaptive nature of BNPO, while theoretically sound, may introduce additional computational overhead in estimating the parameters of the Beta distribution during training. The paper could discuss the trade-off between the benefits of the method and its computational costs more explicitly.
3. The paper compares BNPO primarily with methods like REINFORCE, GRPO, and REINFORCE++. It would be valuable to see comparisons with a broader range of reinforcement learning algorithms or real-world scenarios beyond reasoning tasks.
4. There are no detailed ablation studies presented to demonstrate the individual contributions of different components of BNPO, such as the Beta normalization and advantage decomposition mechanisms. This would provide clearer insights into the importance of each part of the method.
5. The method’s performance on larger, more complex datasets and models is promising, but scalability for extremely large-scale models or highly complex environments needs further investigation.
6.  Although BNPO shows strong results on reasoning tasks, its generalization to other domains or tasks with different reward structures and dynamics could be further tested. This would provide a broader validation of the method’s robustness.

**Questions:**

1. How does BNPO perform when applied to non-binary reward structures, especially in tasks with multi-modal or continuous reward functions? Could there be challenges in adapting the Beta distribution in such cases?
2. Can BNPO maintain its performance and stability with significantly larger models, such as those with billions of parameters, without encountering computational bottlenecks?
3. How do the hyperparameters in BNPO, particularly the Beta distribution parameters (α, β), interact with other optimization hyperparameters like learning rate and batch size? Is there a risk of overfitting or sensitivity to these parameters?

---

> ### Author Response · Authors · 2025-11-25
>
> We appreciate your careful and constructive comments. We have addressed the questions that you raised as follows. Please let us know if you have any further concerns.
>
> $\textbf{Q1:}$ While BNPO works well for binary-valued rewards, the extension to multi-valued or continuous rewards is mentioned but not thoroughly explored in the paper. More detailed analysis and experiments on this extension would strengthen the claim of BNPO's general applicability.
>
> $\textbf{A1:}$ We do not aim to conduct experiments with binary-valued rewards, as DeepSeek-R1 has already demonstrated that such rewards can yield strong RL performance. To evaluate the effectiveness of BNPO under continuous or multi-valued rewards, we introduce an additional length reward [1]. The results, shown in the table below, indicate that BNPO remains effective in this setting.
>
> | Method       | Acc|Length|
> |--------------|--------|--------|
> | REINFORCE    |38.3  | 785  |
> | REINFORCE++  | 38.2  | 784  |
> | GRPO         | 38.3  | 744  |
> | ReMax        | 37.5 | 795  |
> | BNPO         | 38.5  |784  |
>
>
> [1] Yu Q, Zhang Z, Zhu R, et al. Dapo: An open-source llm reinforcement learning system at scale[J]. arXiv preprint arXiv:2503.14476, 2025.
>
> $\textbf{Q2:}$ The adaptive nature of BNPO, while theoretically sound, may introduce additional computational overhead in estimating the parameters of the Beta distribution during training. The paper could discuss the trade-off between the benefits of the method and its computational costs more explicitly.
>
> $\textbf{A2:}$ Since the computation of the advantage function in BNPO involves only scalar operations, it is highly efficient. As a result, the additional computational overhead introduced by BNPO, compared to GRPO and REINFORCE, is negligible. The runtime of differernt policy optimization methods are shown in the following table. The running time of BNPO is about the same as that of GRPO and REINFORCE. Therefore, BNPO does not compromise training efficiency.
>
> | Method       | Time  |
> |--------------|--------|
> | REINFORCE    | 23.9h  |
> | REINFORCE++  | 23.9h  |
> | GRPO         | 24.2h  |
> | ReMax        | 30.4h  |
> | BNPO         | 23.9h  |
>
> $\textbf{Q3:}$ The paper compares BNPO primarily with methods like REINFORCE, GRPO, and REINFORCE++. It would be valuable to see comparisons with a broader range of reinforcement learning algorithms or real-world scenarios beyond reasoning tasks.
>
> $\textbf{A3:}$ Thank you for your suggestion. Due to the property of the LLM, which requires much memory and computation, other optimization methods, such as PPO, may require more memory and computation. We do not select real-world scenarios because these tasks are commonly open-ended. It is difficult to evaluate and fairly compare the performance under this circumstance. We will explore the application of BNPO with broader tasks in future.
>
> $\textbf{Q4:}$ There are no detailed ablation studies presented to demonstrate the individual contributions of different components of BNPO, such as the Beta normalization and advantage decomposition mechanisms. This would provide clearer insights into the importance of each part of the method.
>
> $\textbf{A4:}$ We have presented ablation studies to demonstrate the effectiveness of the Beta Normalization and advantage decomposition mechanisms.
>
> Without Beta Normalization, BNPO reduces to REINFORCE. As shown in Table 1, BNPO achieves better performance compared to REINFORCE, demonstrating the effectiveness of the Beta Normalization.
>
> We did not omit the experimental analysis of the advantage decomposition mechanism. Due to space constraints, it is included in the appendix (Page 13, Paragraph “Advantage Decomposition”). As shown in Table 4, both GRPO with advantage decomposition and REINFORCE++ with advantage decomposition achieve slight improvements over their original versions. BNPO continues to achieve the best average performance among all methods.

---

> ### Author Response · Authors · 2025-11-25
>
> $\textbf{Q5:}$ The method’s performance on larger, more complex datasets and models is promising, but scalability for extremely large-scale models or highly complex environments needs further investigation.
>
> $\textbf{A5:}$ The base model used in our experiments, Qwen2.5-Math-7B, is already a relatively large and widely adopted model, and our method consistently produces improvements over this strong baseline.
>
> Since our study involves training the base model for more than a thousand steps, the computational and runtime costs are substantial. Scaling the same extended training schedule to larger models, such as the Qwen-32B series, would require several weeks of computation and is therefore infeasible within a reasonable timeframe.
>
> $\textbf{Q6:}$ Can BNPO maintain its performance and stability with significantly larger models, such as those with billions of parameters, without encountering computational bottlenecks?
>
> $\textbf{A6:}$ The base model used in our experiments, Qwen2.5-Math-7B, contains over 7 billion parameters and is already a very large model. As shown in Table 1 and Figure 2, BNPO consistently maintains strong performance and stable behavior on this 7B-scale backbone. Moreover, the runtime of BNPO is comparable to that of GRPO and REINFORCE. Consequently, BNPO faces the same computational bottlenecks as GRPO when scaling to even larger models.
>
> $\textbf{Q7:}$ How do the hyperparameters in BNPO, particularly the Beta distribution parameters $(\alpha,\beta)$, interact with other optimization hyperparameters like learning rate and batch size? Is there a risk of overfitting or sensitivity to these parameters?
>
> $\textbf{A7:}$ The computation of the Beta distribution parameters in BNPO is directly based on the expected reward $p$, meaning that no additional learnable parameters are introduced and, therefore, no extra risk of overfitting or sensitivity is incurred. The stable gradient norms observed in Figure 2 further support this property. Moreover, because the BNPO parameters are defined independently of the optimization hyperparameters, the method is inherently robust to changes in other optimization settings.

---

> > ### Comment · Reviewer_xn5Y · 2025-11-25
> >
> > Thanks for your rebuttal. I have no other concerns and decide to maintain my positive score.

---

### Official Review · Reviewer_n46U · 2025-11-01

**Soundness:** 3
**Presentation:** 3
**Contribution:** 2
**Rating:** 4
**Confidence:** 3

**Summary:**

The paper proposes reducing the variance of RL gradients by introducing a novel normalization, normalized by a beta distribution. The parameters of the beta distribution are dynamically computed according to the reward mean and variance, in order to minimize the gradient variance. The modelling unifies the existing methods, including GRPO and REINFORCE, by showing them taking different distribution parameters. Additionally, it extends to multiple-reward problems through an advantage decomposition mechanism, which normalizes each reward, such as format reward and accuracy reward, separately. The algorithm is tested on four math benchmarks, showing comparable performance to baselines.

**Strengths:**

The paper addresses a practical problem aimed at improving the stability of policy gradient by reducing gradient variance. Meanwhile, it provides an elegant theoretical derivation of the algorithm.

**Weaknesses:**

One of the baselines, ReMax, already provides a straightforward method for computing a baseline and stabilizes the training as shown in Figure 2. The proposed method requires the estimation of two parameters, which can introduce extra estimation biases and variances.

The empirical testing is limited to four MATH benchmarks and Qwen base models only. Also, experimental analysis on the advantage decomposition mechanism is missing.

Li, Z., Xu, T., Zhang, Y., Lin, Z., Yu, Y., Sun, R., & Luo, Z. Q. (2023). Remax: A simple, effective, and efficient reinforcement learning method for aligning large language models. arXiv preprint arXiv:2310.10505.

**Questions:**

1. What is the variance and bias of estimating the parameters $\alpha$ and $\beta$? How will it influence the gradient estimator? As shown in Figure 3, the estimation of $\alpha$ is very unstable.

2. Are there any experiments on the advantage decomposition mechanism?

---

> ### Author Response · Authors · 2025-11-25
>
> We appreciate your careful and constructive comments. We have addressed the questions that you raised as follows. Please let us know if you have any further concerns.
>
> $\textbf{Q1:}$ One of the baselines, ReMax, already provides a straightforward method for computing a baseline and stabilizes the training as shown in Figure 2.
>
> $\textbf{A1:}$ ReMax incurs higher computational cost than other policy optimization methods because it requires additional sampling to estimate the baseline. The runtimes of different policy optimization methods are summarized in the table below. As shown, ReMax requires approximately 25% more runtime than the alternatives. In contrast, our BNPO method achieves the best performance and stability (see Table 1 and Figure 2) while maintaining training efficiency comparable to standard baselines.
>
> | Method       | Time  |
> |--------------|--------|
> | REINFORCE    | 23.9h  |
> | REINFORCE++  | 23.9h  |
> | GRPO         | 24.2h  |
> | ReMax        | 30.4h  |
> | BNPO         | 23.9h  |
>
> $\textbf{Q2:}$ The proposed method requires the estimation of two parameters, which can introduce extra estimation biases and variances.
>
> $\textbf{A2:}$ We empirically verify that estimating the two parameters in BNPO introduces a small amont of bias or variance. To evaluate estimation error, we use the average of log-likelihood to measure the goodness of fit. As shown in Figure 6 (Page 14), the value remains within a relatively small range in most steps, indicating that the fitted distribution matches the empirical distribution and introduces a small amount of bias.
>
> Figure 2 further shows that BNPO maintains a stable and low gradient norm throughout training. In addition, Table 3 (Page 12) reports that BNPO attains the lowest gradient variance among all compared methods. Taken together, these results indicate that BNPO not only avoids increasing estimation variance but in fact reduces it.
>
> $\textbf{Q3:}$ What is the variance and bias of estimating the parameters $\alpha$ and $\beta$? How will it influence the gradient estimator? As shown in Figure 3, the estimation of $\alpha$ is very unstable.
>
> $\textbf{A3:}$ The instability of $\alpha$ does not arise from the Beta Normalization itself but from fluctuations in $E[p]$ and $\mathrm{Var}[p]$, because $\alpha$ and $\beta$ are computed directly from these two moments, which are inherently unstable during training.
>
> Since the expected reward $p$ depends on the model capability, which evolves continuously, the distribution of $p$, and consequently $E[p]$ and $\mathrm{Var}[p]$, naturally changes over time. Beta Normalization is introduced precisely to adapt to this dynamic behavior and to mitigate the resulting instability in the distribution of $p$.
>
> If we fix $\alpha$ and $\beta$ to constant values such as 1 or $\frac{3}{2}$ to make them stable, BNPO reduces to REINFORCE or GRPO, respectively. As shown in Figure 2, the gradient norm becomes unstable under these fixed settings and exhibits abrupt fluctuations across many training steps. This happens because the distribution of $p$ changes during training, while fixed values of $\alpha$ and $\beta$ cannot adjust to this change.
>
> Another factor may be visual bias. Since $\alpha$ is numerically larger than $E[p]$, changes in $\alpha$ appear more noticeable in the plots. For clarity, we also provide the curves of $E[p]$ and $\mathrm{Var}[p]$ in Figure 7 and Figure 8 (Page 15) separately, which clearly show that both quantities vary throughout training.
>
> $\textbf{Q4:}$ The empirical testing is limited to four MATH benchmarks and Qwen base models only.
>
> $\textbf{A4:}$ We use the MATH benchmarks because their solutions are easy to verify, and we adopt the Qwen base models due to their strong performance. Both are widely used in prior work.
>
> We additionally report the performance of different policy optimization methods on the GPQA and MMLU-Pro datasets using the Llama-3.1-8B-Instruct model. As shown in the table below, BNPO achieves the best average performance among all compared methods.
>
> | Method       | GPQA|MMLU-Pro|
> |--------------|--------|--------|
> | REINFORCE    |31.2  | 49.3  |
> | REINFORCE++  | 29.7  | 48.2  |
> | GRPO         | 30.0  | 48.1  |
> | ReMax        | 29.6  | 48.4  |
> | BNPO         | 31.4  | 49.8  |
>
> $\textbf{Q5:}$ Experimental analysis on the advantage decomposition mechanism is missing.
>
> $\textbf{A5:}$ We did not omit the experimental analysis of the advantage decomposition mechanism. Due to space constraints, it is included in the appendix (Page 13, Paragraph “Advantage Decomposition”). As shown in Table 4, both GRPO with advantage decomposition and REINFORCE++ with advantage decomposition achieve slight improvements over their original versions. BNPO continues to achieve the best average performance among all methods.

---

### Author Response · Authors · 2025-11-25

We apologize for the slight delay, which allowed us to provide more complete responses and include additional experimental results. Please feel free to reach out if you have any further questions or concerns.

---

### Note · Authors · 2026-01-31

**Comment:**

Dear Reviewer 9sLr:

Thank you for your time and effort in reviewing our submission. However, we were unable to reach an agreement on some points during the discussion. We would like the involvement of other reviewers to provide a fresh perspective.

**Withdrawal Confirmation:**

I have read and agree with the venue's withdrawal policy on behalf of myself and my co-authors.

---

### Meta-Review · Area_Chair_GSaJ · 2026-01-06

**Summary:**

This paper studies reward normalization strategies for REINFORCE-style policy gradient algorithms, with a primary focus on binary reward settings. It introduces Beta Normalization Policy Optimization (BNPO), which normalizes the centered reward $R(q,o)−p(q)$ using the density of a Beta distribution with dynamically estimated parameters, aiming to reduce gradient variance (where $\mathbb{E}[R \rvert q]$ denotes the success probability for question $q$). Under this formulation, BNPO provides a unifying perspective on existing methods such as REINFORCE and GRPO by showing that they correspond to different choices of distribution parameters. The framework is further extended to multi-reward settings via an advantage decomposition scheme, in which individual rewards (e.g., format and accuracy rewards) are normalized separately. Empirical evaluations on four mathematical reasoning benchmarks show that BNPO achieves performance comparable to strong baselines including REINFORCE, GRPO, and REINFORCE++.

The main strengths of this work:

- Reviewers all appreciate the importance and the novelty of the dynamic reward normalization technique given that the existing methods use static normalization, which is not adaptive to the training progress.
- It is interesting that with a different parameter setup, Beta normalization can recover widely used methods like REINFORCE and GRPO.
- BNPO achieves strong results on math reasoning tasks and is comparable to the baselines but with lower gradient variance during training.

At the same time, the reviewers also raised several concerns:

- (1) Theoretical soundness of the normalized gradient estimator (Reviewers 9sLr, xn5Y, and qMA3):

(i) Several concerns focus on whether the proposed normalized gradient estimator remains a meaningful approximation to the true policy gradient, with doubts about unbiasedness and convergence. The variance-reduction argument is challenged for relying on unrealistic independence assumptions, and the modeling of success probabilities with a single Beta distribution is seen as fragile, especially under non-stationarity and policy dependence. (ii) There are also concerns that density-based normalization can amplify noise and lead to unstable training under mis-specified parameters, without sufficient theoretical or empirical justification of when and why it should work.

- (2) More empirical evaluation, ablations, and statistical justification needed (Reviewers n46U, xn5Y, 9sLr, qMA3):

(i) Concerns include limited benchmarks (mostly MATH-style reasoning tasks and Qwen base models), marginal or inconsistent performance gains, and missing evaluations on harder tasks, larger models, or diverse domains.
(ii) Reviewers also repeatedly requested ablation studies to isolate the contribution of Beta normalization and advantage decomposition, as well as diagnostics such as gradient-variance curves, sensitivity to mis-specified parameters, and empirical validation that the success probabilities indeed follow a Beta distribution.
(iii) The lack of confidence intervals, alternative metrics beyond simple averages, and clearer interpretations of reported gains makes the empirical claims less convincing.

- (3) Baselines and related work (Reviewer n46U, xn5Y, qMA3):

(i) Reviewers note that existing baselines such as ReMax already provide a simple and stable variance-reduction mechanism, raising questions about the necessity of BNPO’s additional parameter estimation. (ii) Comparisons are largely restricted to REINFORCE-style methods, and the reviewers requested more comparison with broader baselines (e.g., other RL algorithms and normalization strategies). (iii) The related work is also seen as somewhat narrow, lacking a broader discussion of normalization techniques in RL, including reward and adaptive coefficient normalization.


- (4) Computational cost and scalability (Reviewer xn5Y, qMA3, n46U):

(i) Estimating Beta distribution parameters via Monte Carlo sampling introduces extra computation, raising questions about runtime overhead and scalability to very large models. (ii) Reviewers requested explicit runtime comparisons, a clearer exposition of hyperparameter selection, and a more transparent discussion of cost–benefit trade-offs. Given the marginal performance improvements, several reviewers also questioned whether BNPO offers enough practical advantage to justify its added complexity.

**Reviewer Concerns:**

After the rebuttal, the concerns (2) and (4) have been addressed or alleviated, while concerns (1) and (3) still largely remain. Specifically:

As for (1), regarding the unbiasedness and convergence, the rebuttal clarified that BNPO is designed to optimize an objective function $\mathbb{E}[\tilde{R}(q,o; \alpha,\beta)]$ different from the original one $\mathbb{E}[\tilde{R}(q,o)]$. However, this does not address the reviewer’s question since the optimal policies for the landscape of the two objectives can likely be very different, and using the gradient updates of a surrogate objective function does not necessarily imply convergence under the original objective function. Moreover, $\mathbb{E}[\tilde{R}(q,o; \alpha,\beta)]$ depends on the choices of $\alpha$ and $\beta$ and hence is non-stationary during training. It is not clear what the policy would ultimately converge onto under gradient updates with respect to a non-stationary objective function. Therefore, the concern about the theoretical soundness still remains.

Regarding concern (3): As for (i), in the rebuttal, the authors stated that compared to ReMax, the main strength of BNPO is the training time (about 25% less). However, I do not find this a convincing argument given that ReMax achieves comparable accuracy on all the math datasets (Table 1) and its training stability also appears quite comparable (Figure 2). A more concrete comparison between ReMax and BNPO is needed.
(ii) appears to remain unaddressed. Regarding (iii), some normalization techniques in RL are added and discussed during the rebuttal.

**Reviewer Scores:**

This paper received mixed initial reviews (n46U: 4 / xn5Y: 6 / 9sLr: 2 / qMA3: 6).

During the discussions, Reviewer 9sLr expressed the remaining concern on the theoretical soundness.

Reviewers xn5Y and qMA3 responded that they tend to maintain their original scores given that the changes do not appear enough to justify a rating change despite that their concerns are alleviated.

---

### Decision · Program_Chairs · 2026-01-26

Reject